# TIGD1 Function as a Potential Cuproptosis Regulator Following a Novel Cuproptosis-Related Gene Risk Signature in Colorectal Cancer

**DOI:** 10.3390/cancers15082286

**Published:** 2023-04-13

**Authors:** Zhiwei Wu, Changwei Lin, Fan Zhang, Zhixing Lu, Yaohui Wang, Yang Liu, Zhijiao Zhou, Liang Li, Liying Song

**Affiliations:** 1Department of Health Management, The Third Xiangya Hospital, Central South University, Changsha 410013, China; 2Department of Gastrointestinal Surgery, The Third Xiangya Hospital of Central South University, Changsha 410013, China; 3Department of Gastrointestinal, Hernia and Enterofistula Surgery, People’s Hospital of Guangxi Zhuang Autonomous Region, Nanning 530016, China; 4Department of Pathology, The Third Xiangya Hospital of Central South University, Changsha 410013, China; 5Department of Pharmacy, The Third Xiangya Hospital, Central South University, Tongzipo Road, Changsha 410013, China

**Keywords:** copper, cuproptosis, colorectal cancer, prognostic signature, immune microenvironment, TIGD1

## Abstract

**Simple Summary:**

This study investigates the role of cuproptosis, a type of programmed cell death that depends on copper, in the progression of colorectal cancer (CRC). Using bioinformatics analysis of a large dataset of CRC patient samples, the authors identified seven cuproptosis markers. From this, they developed a predictive model that can predict the prognosis of CRC patients. The study also shows that the gene TIGD1 is involved in regulating cuproptosis in CRC cells. These findings provide new insights into the role of cuproptosis in cancer and may lead to new therapeutic strategies for treating CRC.

**Abstract:**

Cuproptosis is a new form of copper-dependent programmed cell death commonly occurring within the body. There is emerging evidence indicating that cuproptosis has a significant regulatory function in the onset and progression of cancer. However, it is still unclear how cuproptosis regulates cancer and whether other genes are involved in the regulation. Using the TCGA-COAD dataset of 512 samples, we found that seven of ten cuproptosis markers showed prognostic value in colorectal cancer (CRC) using Kaplan–Meier survival analysis. Furthermore, 31 prognostic cuproptosis-related genes were identified using weighted gene co-expression network analysis and univariate Cox analysis. Subsequently, we constructed a 7-PCRG signature using least absolute shrinkage and selection operator (LASSO)–Cox regression analysis. The risk score predicting survival in patients with CRC was evaluated. Two risk groups were classified based on their risk scores. The two groups revealed a significant difference in immune cells, such as B and T cells. Furthermore, we identified differences in many immune functions and checkpoints, including CD276 and CD28. In vitro experiments showed that a hub cuproptosis-related gene, TIGD1, could significantly regulate cuproptosis in CRC after exposure to elesclomol. This study validated that cuproptosis was closely related to the progression of CRC. Seven new cuproptosis-related genes were identified, and the function of TIGD1 in cuproptosis was preliminarily understood. Since a certain concentration of copper in CRC cells is important, cuproptosis may provide a new target for cancer therapy. This study may provide novel insights into the treatment of CRC.

## 1. Introduction

Colorectal cancer (CRC) is among the most prevalent cancer types worldwide and is the third leading cause of cancer-related death [1,2]. Although the overall survival (OS) rate of patients with early-stage CRC has tremendously improved over the last few decades, their clinical prognosis remains far from satisfactory because of late diagnosis, rapid progression, and early metastasis [3,4].

The roles of many cell-death patterns in cancer, such as apoptosis, autophagy, and necrosis, have been extensively studied, and they have been significantly associated with CRC [5,6,7,8]. Recently, a series of newly emerging regulated cell death pathways, including pyroptosis, necroptosis, and ferroptosis, have been repeatedly studied for their close correlations with the development of many cancers [8,9,10,11]. Many authentic studies have revealed that the induction of these patterns combined with immune checkpoint inhibitors synergistically enhanced antitumor activity, indicating they may play important roles in antitumor immunity [12,13]. Therefore, identifying more patterns of death will definitely expand our views and strategies for cancer treatment. 

Cuproptosis is a recently discovered copper-dependent cell death pathway [14]. A recent study by Tsvetkov P et al. revealed that excessive copper accumulation in cells triggers the aggregation of mitochondrial lipoylated proteins and causes Fe-S cluster proteins to become unstable. They coined a new term to describe this: cuproptosis [15]. They also reported that cuproptosis is completely different from traditionally reported cell death caused by oxidative stress (e.g., apoptosis, ferroptosis, and necroptosis) and can be initiated via an independent pathway. Considerable evidence suggests that copper is an important factor remarkably influencing many biological processes in cancer, including lipolysis, autophagy, and cell growth [16,17]. Therefore, it is wise to conclude that cuproptosis may harbor the potential for treating cancer. However, the mechanisms underlying the regulation of cancer by this newly defined cell death need to be further clarified. Previous studies have identified several novel genes involved in the processes of ferroptosis and necroptosis. However, only a few studies have reported experimental evidence on new genes that regulate cuproptosis in cancers. Limited research has been conducted on the relationship between cuproptosis and CRC. Therefore, the cuproptosis-related regulatory network requires further study. Therefore, we have reason to hypothesize that there must be more novel genes involved in the process of cuproptosis in CRC. In addition, understanding how they participate in the regulation of cuproptosis in CRC is unquestionably important. 

In this study, we comprehensively evaluated the overall landscape of cuproptosis regulation in CRC. Subsequently, we used weighted gene co-expression network analysis (WGCNA) to identify a hub module with 547 mRNAs that are closely correlated to the progression of cuproptosis in CRC. More importantly, we conducted least absolute shrinkage and selection operator (LASSO)–Cox regression analysis to further filter out seven novel genes and establish a cuproptosis-related mRNA prognostic signature. Further investigation revealed that TIGD1, a hub gene in our signature, could regulate cuproptosis in CRC.

Our study provides novel insights on CRC diagnosis and treatment from a cuproptosis-related perspective. The establishment of this signature and the identification of TIGD1 in cuproptosis regulation may ultimately translate into better clinical guidance for CRC patients. By providing a more accurate prognosis and guiding treatment selection, this study has the potential to enhance treatment sensitivity and improve the survival rates of CRC patients.

## 2. Materials and Methods

### 2.1. Data Acquisition

The RNA-seq data of 512 CRC samples, including 41 normal samples, and 473 tumor samples and their corresponding clinical characteristics were downloaded from The Cancer Genome Atlas (TCGA) website (https://portal.gdc.cancer.gov/projects/TCGA-COAD, accessed on 13 June 2022). Subsequently, Ensembl gene IDs were converted to official gene symbols using GENCODE.v22, and log2 processing of the data was performed.

All protein-coding genes were screened using the Ensembl human genome browser, GRCh38. Data normalization was then conducted by converting raw counts into fragments per kilobase of exon model per million mapped fragments (FPKM).

GSE17538 and 238 CRC samples were downloaded from the Gene Expression Omnibus Database (https://www.ncbi.nlm.nih.gov/geo/query/acc.cgi, accessed on 23 August 2022). Next, the IDs were annotated using the GPL570 platform.

### 2.2. Differential Expression Analysis

Differential expression analyses were conducted according to previous studies [18]. Briefly, the “limma” package [19] was used to screen mRNA expression matrices between COAD and normal samples. Multiple testing correction is necessary to account for the multiple hypothesis tests performed during differential expression analysis. In this study, we used the false discovery rate (FDR) correction for adjustments. The criteria for DEmRNAs were |log 2(fold change)| > 1 and FDR < 0.05.

### 2.3. Identification of Cuproptosis-Related Genes by WGCNA

The 10-marker list was obtained from the latest research. Next, WGCNA was applied to identify core modules and hub genes related to these markers using the “WGCNA” R package [20]. The identified differentially expressed genes (DEGs) between normal and tumor tissues in COAD were used to construct a scale-free network. The Pearson correlation coefficients between each genetic module were extracted to establish the module–trait relationship between the expression levels of DEGs and cuproptosis-related genes according to the β-value (soft threshold). The modules that showed the highest correlation with all cuproptosis-related genes were selected for further research. Gene significance (GS) represented the level of correlation between DEG expression and traits and was generated using linear regression. Module significance (MS) was defined as the average GS of all genes involved in the module. The inclusion criteria for hub genes in the module were as follows: (module membership) MM > 0.8 and GS > 0.2 [21].

### 2.4. Construction of the Cuproptosis-Related Prognostic Signature

We constructed the signature using the method described in our previous study [21]. The candidate genes filtered out in the core module of WGCNA were filtered using Cox univariate analysis and the “survival” R package. All patients with CRC were randomly separated into training or testing cohorts in the ratio of 1:1. Subsequently, LASSO–Cox regression analysis was used to evaluate these prognostic candidate genes. By selecting the optimal penalty parameter correlating with the minimum 10-fold cross-validation, we established a seven-gene optimal prognostic model. The formula for cuproptosis-related prognostic risk scores for each patient was
$$\mathrm{Risk}\;\mathrm{score}=\sum_{i}^{n}Coefi\times Xi$$
where *Xi* and *Coefi* respectively represent the expression of each gene and its corresponding coefficient. The patients in the training cohort were then divided into low-risk and high-risk groups according to the median risk score. We constructed Kaplan–Meier curves using the “survminer” R package with the log-rank test to compare the OS rates between the risk groups. The log-rank test assumes that the hazard ratios are constant over time and that the survival curves do not cross. The test statistic is based on the difference between the observed and expected number of events in each group, and it follows a chi-squared distribution with one degree of freedom. A receiver operating characteristic curve (ROC) [22] was also generated to evaluate the OS predictive value of the novel signature using the “timeROC” R package. To assess the feasibility of the model, we calculated the risk score of the testing cohort based on the same formula used for the training cohort and then performed the same validation analyses as mentioned above.

### 2.5. Gene Set Enrichment Analyses

To identify differential signaling pathways between the two risk groups, gene set enrichment analyses (GSEA) were performed using GSEA software 4.0.1. The enrichment levels and statistical significance were determined through normalized enrichment scores and nominal *p*-values. Furthermore, single-sample GSEA (ssGSEA) was conducted using the ‘GSVA’ R package based on a previous study [23] to analyze the infiltrating scores of 16 immune cells and the activities of 13 immune-related pathways between the two risk groups.

### 2.6. Functional Enrichment Analysis

For the genes in the high-risk and low-risk groups, the “clusterProfiler” package [24] was used to perform functional enrichment in the Gene Ontology (GO) and Kyoto Encyclopedia of Genes and Genomes (KEGG) databases. KEGG is a database that helps researchers understand high-level biological functions and contains pathway maps and associated metabolic and signaling pathways. GO is a standardized vocabulary that describes the attributes of genes and gene products across all organisms in terms of their biological processes (BP), cellular components (CC), and molecular functions (MF). It provides a basic framework for subsequent functional analysis.

During functional enrichment analysis, differentially expressed genes were selected and compared to the terms in either the Gene Ontology or KEGG databases. The number of genes that matched each term was determined, and hypergeometric tests were used to identify significantly enriched entries. An adjusted *p*-value threshold of less than 0.05 was used to determine statistical significance.

### 2.7. Assessment of Immune Cell Infiltration and Immune Microenvironment

The ESTIMATE algorithm was used to assess immune infiltration in patients with CRC [25]. The differences in immune cell infiltration between the two groups were evaluated using the CIBERSORT algorithm. CIBERSORT is an analysis tool that uses expression data to calculate the cell composition of complex tissues based on preprocessed gene expression profiles [26]. LM22 of CIBERSORT defines 22 immune cell subsets obtained from the CIBERSORT web portal (http://CIBERSORT.stanford.edu/, accessed on 13 June 2022). Finally, Tumor Immune Dysfunction and Exclusion (TIDE; http://tide.dfci.harvard.edu/, accessed on 13 June 2022) algorithms were used to predict the immune checkpoint response to immunotherapy. *p* < 0.05 was considered statistically significant [27].

### 2.8. Drug Sensitivity Prediction

Drug sensitivity prediction was analyzed based on a previous study [21]. The “pRRophetic” [28] R package was used to predict the IC_50_ of chemotherapeutic drugs; this value indicates the effectiveness of a substance in inhibiting specific biological or biochemical processes.

### 2.9. Tissue Sample Collection, RNA Extraction, and Quantitative Real-Time Polymerase Chain Reaction (qRT-PCR)

All samples were retrieved according to previously published methods [21]. Briefly, tissue samples were collected from our hospital, and the study was approved by the Medical Ethics Committee of the hospital. Ten pairs of clinical samples, including tumor and pericardial tissues, were obtained from patients with CRC who underwent tumor resection surgery between October 2020 and August 2021. All samples were stored in an −80 °C freezer before use.

RNA was extracted from tissues using the SteadyPure Universal RNA Extraction Kit (AG21017, Accurate Bio) based on standard protocols. Subsequently, cDNA was synthesized using the obtained RNAs and an Evo M-MLV RT Kit (AG11707, Accurate Bio, Hunan, China). Genetic expression levels were quantified using a Roche LightCycler 480 with SYBR Green Master Mix (11201ES03, Yeasen, Shanghai, China), and the expression levels were calculated using the 2^−ΔΔCt^ method. GAPDH served as an internal reference for normalization. All primers used for qRT-PCR were synthesized by Tsingke Biotech (Tsingke, Beijing, China). The primer sequences used are listed in Appendix A.

### 2.10. Cell Viability and Colony Formation Assays

A total of 5000 cells were plated into a 96-well plate and incubated for 24 h. The cells were then exposed to drugs for 24 h. Subsequently, 100 μL of fresh medium containing 10% cell counting Kit-8 (CCK8) solution (MA0218, Meilunbio, Dalian, China) was added. The number of cells was determined by measuring the absorbance at 450 nm using a 96-well plate reader (PerkinElmer, Hopkinton, MA, USA). For the colony formation assay, 500 cells were seeded into each well of a 6-well plate in triplicates under different conditions and incubated for 14 days. Colonies were fixed with paraformaldehyde and stained with crystal violet. Average colony counts were calculated, and a paired t-test was used to evaluate statistical significance.

### 2.11. Chemical Reagent Assay

All chemical reagents (elesclomol (T6170), Q-VD-Oph (T0282), ferrostatin-1 (T6500), necrostatin-1 (T1847), and 3-methyladenine (T1879)) were purchased from Topscience (Shanghai, China).

A total of 5000 cells were plated into a 96-well plate and incubated for 24 h. Subsequently, appropriate inhibitors were added for 12 h. The cells were then exposed to Elesclomol and Cucl2 for 24 h. In addition, 100 μL of fresh medium containing 10% cell counting Kit-8 (CCK8) solution (MA0218, Meilunbio, Dalian, China) was added. The number of cells was determined by measuring the absorbance at 450 nm using a 96-well plate reader (PerkinElmer, Hopkinton, MA, USA).

### 2.12. Western Blotting

Western blotting was performed following the official protocol. In brief, protein was extracted from the cells and tissues via treatment with 1× radioimmunoprecipitation assay buffer (KeyGEN, Nanjing, China) containing 1% phenylmethylsulphonyl fluoride (KeyGEN). The proteins were separated by sodium dodecyl sulfate-polyacrylamide gel electrophoresis, transferred to polyvinylidene fluoride membranes (Millipore, Bedford, MA, USA), blocked with 5% skim milk for 1 h at room temperature, immunoblotted overnight with primary antibodies and for 1.5 h with secondary antibodies, and visualized using an Odyssey CLx Infrared Imaging System (LI-COR Biosciences, Lincoln, NE, USA). Antibodies against TIGD1 were purchased from Wanlei Technology, and antibodies against β-actin were purchased from Proteintech.

### 2.13. Detection of Viable and Dead Cells Using Calcein-Methyl 4-Acetoxybenzoate (Calcein-AM)/Prodium Iodide (PI)

The viable and dead cells were detected using a calcein-AM/PI kit (YEASEN, Shanghai, China). Briefly, RKO cells were collected and centrifuged; the supernatant was then discarded. After rinsing with assay buffer three times, the 100 µL of staining reagent was added to the cell mixture and incubated for 15 min at 37 °C.

### 2.14. Measurement of the Copper Content

CRC cells were plated in 6-cm plates overnight and treated with elesclomol for 24 h. The cells were then collected and resuspended in 100 μL of double-distilled water. They were ultrasonically disrupted to obtain intracellular copper for detection according to the manufacturer’s protocol (E-BC-K775-M, Elabscience, Wuhan, China).

### 2.15. Statistical Analysis

Bioinformatic analyses based on the R language were used to conduct statistical analyses. Statistical significance between the two groups was calculated using the Student’s t-test. Differences between the groups were analyzed using one-way ANOVA. The Mann-Whitney test using the BH method adjusted *p*-value was adopted to measure the ssGSEA scores. Statistical significance is defined as *p* < 0.05.

## 3. Results

### 3.1. Expression Landscape of Cuproptosis Markers in CRC

The research flowchart of our study is shown in Figure 1. The data for 512 CRC samples were downloaded from the TCGA database (https://portal.gdc.cancer.gov/repository, accessed on 13 June 2022). To illustrate the expression differences among these ten markers in CRC, we compared their mRNA expression levels in the TCGA-COAD cohort. The results revealed that eight genes (not *LIAS* or *PDHA1*) were differentially expressed in CRC tissues compared with normal tissues (Figure 2A). Among these *DEGs*, *FDX1*, *DLD*, *DLAT*, *PDHB*, and *MTF1* were downregulated in CRC, whereas the remaining three genes (*LIPT1*, *GLS*, and *CDKN2A*) were upregulated in CRC (Figure 2B). To evaluate the prognostic values of these markers, we explored the relationships between these ten genes and the patients’ OS rates using Kaplan–Meier survival analysis. We found that *FDX1*, *LIAS*, *PDHB*, *DLD*, and *DLAT* were “protective” genes, whereas *CDKN2A* and *LIPT1* were considered “risk” genes (Figure 2C). We subsequently identified the correlations between the ten cuproptosis markers, and they were mostly found to be positively correlated. It is interesting that CDKN2A was negatively correlated with all other genes (Figure 2D). In conclusion, these data suggest that certain cuproptosis markers are differentially expressed in CRC and are correlated with the prognosis of patients with CRC.

### 3.2. Relationship between Cuproptosis Marker Expression Levels and Clinicopathological Features of Patients with CRC

We comprehensively analyzed the relationships between the ten cuproptosis markers and the clinical features of patients with CRC, including MSI status, venous invasion situation, and tumor stage (T stage, N stage, and M stage). The results indicated that many of these markers (*DLD*, *DLAT*, *PDHA1*, *PDHB* and *CDKN2A*) were highly correlated with lymphatic metastasis (Appendix A). *DLD*, *PDHB*, and *GLS* might be correlated with distant metastasis (Appendix A). However, none of these genes were correlated with the depth of tumor infiltration (Appendix A). Four markers were present at low levels in patients with advanced CRC (Appendix A). Moreover, MTF1 might be associated with the MSI status of patients with CRC (Appendix A), and *CDKN2A* was significantly upregulated in patients with CRC exhibiting venous invasion (Appendix A). These data suggest that the abovementioned cuproptosis markers are intrinsically connected and significantly involved in processes affecting CRC progression.

### 3.3. Detection of Cuproptosis-Related mRNAs Using WGCNA

We used WGCNA to identify novel cuproptosis-related mRNAs in CRC. When the scale-free topology model fit reached >0.8, the soft-thresholding power was 7 (β = 7) (Figure 3A). We accordingly confirmed seven mRNA co-expression modules and evaluated their associations with ten cuproptosis markers (Figure 3B). The blue modules (547 mRNAs) were significantly correlated with all markers compared with the other modules (Figure 3C,D). All markers except CDKN2A were positively correlated with the blue module. Among these nine positively correlated genes, GLS, DLAT, and DLD shared high correlation coefficients (R = 0.63, 0.86, and 0.72, respectively) with the blue module (Figure 3E–G). The relationships between the remaining seven markers and the blue module are shown in Appendix A. Thus, we identified a hub module with 547 mRNAs (Appendix A) that are closely correlated to the progression of cuproptosis in CRC.

### 3.4. Construction and Validation of the Prognostic Cuproptosis-Related Gene Signature

Using prognostic information acquired from the TCGA database, univariate Cox regression analyses were used to identify prognostic mRNAs in CRC. We found a total of 2044 mRNAs with prognostic values in CRC patients. Subsequently, 31 prognostic cuproptosis-related genes (PCRGs) were identified by merging the mRNAs in the blue module with the prognostic gene list of CRC (Figure 4A). To identify the potential prognostic prediction power of these PCRGs, all samples were randomly classified into two cohorts: a training cohort and an internal testing cohort. GSE17538 was used as the external testing cohort. The relevant clinical characteristics of patients with CRC in the two cohorts are shown in Table 1.

We performed a LASSO–Cox analysis based on 31 PCRGs and generated a cuproptosis-related gene signature containing seven PCRGs using the training cohort data (Figure 4B,C). We then evaluated the prognostic value of this 7-PCRG signature. According to the coefficient of each PCRG, we calculated a risk score for each patient based on the signature algorithm, and the patients were accordingly classified into low-risk and high-risk groups. A Sankey diagram was plotted to demonstrate the degree of correlation between different risk groups and live statuses (Figure 4D). Cox univariate and multivariate regression analyses were applied to test the independent predictive ability of the signature. Cox univariate regression analysis showed that the risk score of this signature was negatively correlated with the prognoses of patients with CRC (hazard ratio, HR = 2.632; *p* < 0.001; Figure 4E). Furthermore, multivariate Cox regression analysis revealed that only our signature could act as an outstanding independent prognostic factor for predicting the prognosis of patients with CRC (HR = 2.370 and 1.034; *p* < 0.001; Figure 4F). A predictive nomogram was plotted to calculate the possibility of survival for patients with CRC by summing up the scores assigned to many clinical features on a point-based scale along with our signature. We found that the OS rates of patients with CRC were accurately predicted when compared with the prediction ability of the ideal predictive model (Appendix A).

Subsequently, the distribution of the risk scores (above) and the distribution of OS rates (below) were demonstrated to indicate that patients with CRC were reasonably distributed between the high-risk and low-risk cohorts (Figure 5A–C). Kaplan–Meier survival curves were plotted to show that patients with CRC in the low-risk group had superior OS rates than those in the high-risk group in the training (Figure 5D), internal testing (Figure 5E), and external testing (Figure 5F) cohorts. A time-dependent ROC curve was also generated to validate the predictive effect of the cuproptosis-related gene signature. The areas under the curve (AUCs) were maintained at >0.65 at 1, 3, and 5 years in the training and internal testing cohorts (Figure 5G,H). AUCs were also maintained at >0.59 at 1, 3, and 5 years in the external testing cohort (Figure 5I). These results reveal that the prognosis-predictive accuracy of this signature was found to be robust in the training and testing cohorts.

### 3.5. Relationship between the 7-PCRG Signature and the Clinicopathological Characteristics of Patients with CRC

We initially identified the clinicopathological differences between the two risk groups. Significant differences were observed in the tumor stage (*p* < 0.01), T stage (*p* < 0.05), N stage (*p* < 0.001), M stage (*p* < 0.01), microsatellite stability (*p* < 0.001), and during lymph invasion (*p* < 0.01). On further examination of the potential relationship between these PCRGs and clinicopathological characteristics, we noted that six PCRGs with this signature were upregulated in the high-risk group, whereas only SUCLG2 showed an opposite expression tendency (Figure 6A) between the two risk groups. The differences between the clinical characteristics were also compared independently (Figure 6B–E). Altogether, these results suggest that this 7-PCRG signature can ideally predict the tumor stage and prognosis of patients with CRC.

### 3.6. Discovery of Pathways and Molecular Functions of a Cuproptosis-Related Gene Signature via Enrichment Analysis

To explore the underlying difference in signaling pathways or molecular functions between the two risk groups, gene set enrichment analysis (GSEA) was used. Many cancer proliferation pathways were activated in patients in the high-risk group, such as epithelial–mesenchymal transition and hedgehog signaling (Figure 7A). In contrast, certain metabolic pathways, such as butanoate metabolism, fatty acid metabolism, and retinol metabolism, were suppressed in patients in the high-risk group (Figure 7B). We identified DEGs between the two signature-classified risk groups, and using these DEGs, we performed annotation KEGG pathway analysis and GO enrichment analysis (*p* < 0.05). KEGG enrichment revealed that many metastasis-related pathways were significantly enriched in the high-risk group, including focal adhesion, the PI3K-AKT pathway, the MAPK pathway, and the NF-kappa B pathway (Figure 7C). GO enrichment analysis was used to indicate the potential involvement of BP, CC, and MF in CRC. The results are demonstrated in Figure 7D. Several molecular processes related to cancer progression were found to be enriched in BP, CC, and MF. These data further illustrated that cuproptosis may regulate CRC progression via cancer-related signaling pathways such as PI3K-AKT, MAPK, and NF-kappa B pathways.

### 3.7. Immune Landscape of Patients with CRC Using the Cuproptosis-Related Gene Signature

Several studies have validated that ferroptosis, necroptosis, and pyroptosis are involved in immuno-oncology and cancer immunotherapy. Therefore, it is essential to explore whether cuproptosis can influence the cancer immunity of CRC. We also investigated the immune cell infiltration landscape of patients with CRC using the CIBERSORT algorithm. The relative proportion of immune cells is depicted in Figure 8A.

Differences in typical immune cells between the high- and low-risk groups were identified. Furthermore, eight types of immune cells, including naive/memory B cells, plasma cells, CD4 memory/regulatory T cells, M0/M2 macrophages, and eosinophils, were observed with statistically significant differences (Figure 8B). Therefore, cuproptosis may regulate CRC by influencing immune cells.

We also compared the correlation between the risk groups and immune status. Using ssGSEA, we noted that the components of antigen presentation, including the numbers of dendritic cells (DCs), immature dendritic cells (iDCs), co-stimulation of antigen-presenting cells, T-cell co-inhibition, and many immune cells, were significantly different between the patients with low-risk CRC and those with high-risk CRC (Figure 8C). We also observed that a lower level of immune status may result in worse survival (Figure 8D–K). The immune ability was relatively inhibited in the high-risk group. Thus, we can reasonably conclude that immunologic suppression or evasion in high-risk groups may lead to a suboptimal prognosis in patients with CRC.

### 3.8. Immunotherapy and Drug Sensitivity Analysis Based on the 7-PCRG Signature

To further evaluate the sensitivity of immune checkpoint blockades (ICBs) in patients with CRC, we investigated potential changes in immune checkpoint expression in the high-risk and low-risk groups. The expression levels of ICOS, tumor necrosis factor receptor superfamily (TNFRSF)18, TNFRSF25, CD244, CD276, and other genes significantly differed between the two patient groups (Figure 8L). Most checkpoint genes, except TNFRSF25, TNFRSF4, TNFRSF14, TNFRSF18, and CD276, were downregulated in the high-risk group, and therefore, targeted therapy with these five genes might prove ideal for patients in the high-risk group.

Furthermore, TIDE analysis was applied to predict the response of patients with CRC to immune therapy. Patients in the high-risk group had a higher TIDE score, indicating that they harbored a high potential for tumor evasion and were less likely to respond to ICB (Figure 8M). A higher portion of T-cell dysfunction and exclusion was noted in patients in the high-risk group (Figure 8N,O), suggesting that cytotoxic T lymphocytes might be partially disabled in the high-risk group.

Thus, these data demonstrate obvious immunological evasion in high-risk patients with CRC and suggest a potential relationship between cuproptosis and tumor immunity. We also performed a drug sensitivity test to screen for potential drugs to treat patients with non-ideal drug responses. The results revealed that several traditional chemotherapeutic drugs, such as cisplatin (Appendix A) and docetaxel (Appendix A), were not ideal for high-risk patients. However, sunitinib (Appendix A) and erlotinib (Appendix A) might be good drug candidates. The IC_50_ predictions of more representative drugs are shown in Appendix A.

### 3.9. Validation of the Expression Levels of Seven PCRGs in CRC Samples

We further evaluated the expression levels of these seven PCRGs in sample pairs from patients with CRC who underwent treatment at our hospital. As per our data, we observed similar expression trends in the clinical samples (Figure 9A–G). TIGD1 expression levels were remarkably different in tumor tissue and pericardial tissue. Only SUCLG2 showed lower expression levels in tumor tissues than in pericardial tissues; DMPK levels did not differ between tumor and pericardial tissues. The remaining four PCRGs (ASPDH1, PPP1R13L, PMAIP1, and KLHL35) showed higher expression levels in tumor tissues than in pericardial tissues.

### 3.10. Elesclomol-Induced Cuproptosis and the Identification of TIGD1 in Cuproptosis Regulation in CRC

TIGD1 exhibits a high molecular weight in its signature, and its expression levels are remarkably different between CRC and pericardial tissues. We further explored its potential in cuproptosis regulation. Compared with fetal human colon (FHC) cells, the expression level of TIGD1 was significantly upregulated in CRC cell lines, especially in HCT116 cells (Figure 10A,B). Therefore, we induced the stable knockdown of TIGD1 using shRNA in HCT116 cells (Figure 10C,D). The expression level of TIGD1 in shTIGD1#2 was found to be the lowest; therefore, shTIGD1#2 was selected for follow-up research. A previous study reported that elesclomol can act as a copper ionophore and induce cuproptosis in cancer cells [15]. Accordingly, we evaluated whether elesclomol could induce cuproptosis in CRC cells. The CCK-8 assay indicated that the viability of HCT116 cells was inhibited on co-incubation with elesclomol and Cucl_2_, whereas neither standalone elesclomol nor standalone Cucl_2_ could blemish the viability of HCT116 cells (Figure 10E). Furthermore, the copper chelator tetrathiomolybdate liberated the viability of HCT116 cells on incubation with elesclomol; other cell-death inhibitors, including Q-VD-Oph, ferrostatin-1, necrostatin-1, and 3-methyladenine, failed to reverse cell death. These results indicate that elesclomol and Cucl_2_ could successfully induce cuproptosis. Subsequently, after the exogenous administration of elesclomol and Cucl_2_, we examined the intracellular Cu levels and found that they statistically increased in the shTIGD1 group (Figure 10F). Furthermore, on incubation with elesclomol and Cu^2+^, the growth and colony-formation capacity of HCT116 cells could be significantly inhibited in the shTIGD1 group (Figure 10G). The calcein-AM/PI assay also revealed similar results, indicating that shTIGD1 could promote cuproptosis (Figure 10H). Altogether, these data suggest that TIGD1 knockdown might further promote cuproptosis in CRC cells.

## 4. Discussion

The treatment of CRC is a severe clinical predicament owing to its advanced stages and poor prognosis [29]. The molecular identification of diagnostic biomarkers for CRC and its susceptibility to immunotherapy via diagnostic judgment should always be prioritized in scientific research. Extensive research has currently revealed many newly identified cell-death patterns, such as ferroptosis and pyroptosis, which have selective cytotoxic effects against CRC [30,31]. Combined treatment employing these patterns with traditional chemotherapy may enhance the CRC prognosis [32,33]. Many immunotherapies may also restrain tumor cell growth by regulating ferroptosis or pyroptosis [34,35].

Cuproptosis is different from all other types of cell death [12]. Previous studies have demonstrated that excessive copper ions and an increase in copper ionophores can effectively induce cell death [36,37]. Because cancer cells require higher amounts of metals, such as iron and copper, than normal cells to promote rapid metabolism, cuproptosis may act as an immunosuppressor by potentially treating the tumor and playing a role in combination therapy. However, only a few studies have expounded the roles of copper metabolism [38] and copper homeostasis [39] in cancer cells. It was believed that excess copper killed cells by catalyzing the generation of toxic reactive oxygen species. No study has reported the involvement of cuproptosis in tumor immunity in CRC.

We aimed to explore the impact of ten novel cuproptosis markers on CRC prognosis and its clinical characteristics, reveal cuproptosis-related genes, and investigate their potential relationships with tumor immunity. Except for LIAS and PDHA1, the expression levels of the remaining eight markers were statistically different between tumor and normal samples. More importantly, we discovered that FDX1, DLD, and MTF1 were markedly downregulated in CRC. This leads us to infer that their function in CRC may be related to inhibiting cancer progression. In contrast, GLS and CDKN2A showed significantly upregulated expression levels in CRC patients. This suggests that these two genes could serve as potential clinical diagnostic markers for CRC. Furthermore, they might increase the malignancy of CRC by promoting its tumorigenesis and development. Kaplan–Meier analysis revealed that seven of these markers significantly correlated with the OS rates of patients with CRC, and five of them could be considered “protective” genes, whereas CDKN2A and LIPT1 could be considered “risk” genes. The genes are also closely connected, and some of them (such as DLAT and CDKN2A) showed clinically predicted values because they were correlated with clinicopathological features. Some of these markers have been identified for their potential functions in cancer: FDX1 can be activated by SF1 and cJUN in Leydig cells [40], CDKN2a usually functions as a tumor suppressor in many cancers [41,42], and DLAT has also been reported to be an oncogene in gastric cancer [43]. Previous studies also indicate that some of these genes play important roles in CRC. For example, FDX1 expression was associated with quiescence and inflammation but negatively correlated with invasion in colon cancer [44]. CDKN2A, a cell cycle-associated protein, has been identified as promoting CRC metastasis by inducing epithelial-mesenchymal transition [45]. The phosphorylation of PDHA could enhance aerobic glycolysis in CRC cells [46]. GLS is up-regulated in many cancers, and a recent study also showed that GLS depletion could inhibit CRC proliferation and migration through Nrf2 and an autophagy-dependent pathway. Nonetheless, the functions of these markers in CRC progression still need deeper investigation. Moreover, insufficient data are available to explore their roles in cuproptosis.

We comprehensively explored the prognostic values of these cuproptosis markers and suggested that they might significantly influence the development and progression of CRC. Therefore, cuproptosis might be associated with the immune environment of CRC.

The identification of more cuproptosis-related genes may provide novel insights on cuproptosis research; WGCNA is an efficient gene screening method that utilizes transcriptome expression matrices. This bioinformatics algorithm constructs a scale-free network to cluster genes with similar expression patterns, forming different gene modules. In this study, we employed eigengene network methodology to link these modules to 10 identified cuproptosis markers, and we identified a core module (blue) containing 537 novel cuproptosis-related mRNAs. Subsequently, using univariate COX analysis, we narrowed down the list and screened 31 prognostic cuproptosis-related genes. Seven of these were selected to generate a cuproptosis-related gene signature. We classified the patients into two groups based on their risk scores, and several high-risk patients with CRC showed a worse prognosis with distinct advanced clinicopathological stages. Some genes in our signature have been reported to potentially regulate CRC. For example, the PPP1R13 L rs1970764 variant is a potential prognostic marker for patients with rectal cancer [47], and the activation of PMAIP1 induces apoptosis in CRC [48]. However, none of these PRRGs have been correlated with cuproptosis or copper metabolism. We assume that the genes identified in our study might be associated with cuproptosis.

Previous studies have validated that ferroptosis, necroptosis, and pyroptosis are involved in immuno-oncology. We hypothesize that cuproptosis is closely related to tumor immunity. CD8+ T cells have been reported to induce ferroptosis in tumor cells [34], and natural killer (NK) cells and cytotoxic T lymphocytes inhibit tumor cells via pyroptosis [49]. To understand the possible immune landscape of cuproptosis in CRC, we calculated the proportions of different tumor-infiltrating immune cells in CRC using CIBERSORT. Patients in the high-risk group showed relatively downregulated levels of many functional immune cells, such as B cells, plasma cells, CD4 memory T cells, M2 macrophages, eosinophils, and immature immune cells, including M0 macrophages. The levels of these immune cells were significantly related to the progress of patients with CRC [50,51,52]. We can reasonably conclude that the immunological evasion in high-risk patients may lead to suboptimal prognoses in patients with CRC. Alternatively, cuproptosis could be induced in these functional immune cells. Similar findings were noted when investigating immunological states using ssGSEA. We evaluated the enrichment scores of 16 types of immune cells and 13 immune-related functions between the two risk groups. Several immune cells, such as B cells, DCs, iDCs, mast cells, Th2 cells, Treg cells, and NK cells, were found to be downregulated in the high-risk group. Many immune processes were inhibited in the high-risk group, and survival analysis further revealed that the proliferation of immune cells was restrained along with immune-related functions, leading to worse OS rates. We identified obvious immune inhibition in the high-risk group and wondered if ICB therapy would help those patients.

TIDE analysis showed that high-risk patients with CRC have a higher potential for tumor evasion and are less likely to respond to ICBs. The expression profiles of cancer-related checkpoints showed that the majority of checkpoint-target genes had low expression levels in the high-risk group, and only TNFRSF25, TNFRSF4, TNFRSF14, TNFRSF18, and CD276 were upregulated in the high-risk group. The TNFRSF provides crucial costimulatory signals to many immune effector cells and has been identified as a prominent costimulatory domain in CAR-T-cell therapy [53]. Anti-CD276 antibodies eliminate cancer stem cells in a CD8+ T-cell-dependent manner [54]. Therefore, we assumed that targeting TNFSRSF or CD276 may prove ideal for treating high-risk patients with CRC, improving their prognoses by enhancing their immunoreactivity or inducing cuproptosis. Drug sensitivity prediction was analyzed, and many traditional chemotherapy drugs for CRC (cisplatin or docetaxel) were found to be relatively inactive in patients in the high-risk group. This can also partially explain the non-ideal OS rates of high-risk patients. Sunitinib and erlotinib may be useful alternatives for these high-risk patients with CRC, both of which are protein receptor tyrosine kinase inhibitors [55].

Some previous studies have developed cuproptosis signatures for predicting the prognosis of CRC patients [56,57,58,59]. However, these signatures have not undergone external validation or been supported by any cuproptosis-related experiments. Therefore, deeper investigation is required to fully evaluate the role of cuproptosis in CRC.

In our study, TIGD1, a hub gene in our signature, was selected for experimental validation. We noted that TIGD1 knockdown could enhance cuproptosis-induced cell death by increasing the levels of Cu in CRC cells, thereby revealing more novel targets of cuproptosis in CRC. However, the specific mechanisms underlying these effects require further validation.

In summary, we identified seven novel prognostic cuproptosis-related genes that may play a role in regulating cuproptosis in CRC. Additionally, we also validated the function of TIGD1 in CRC. We believe the identification of these novel genes represents a promising avenue for improving early detection, diagnosis, and therapy development for this disease. For instance, these genes may serve as biomarkers for detecting CRC at an early stage. Just like the clinical application targeting ferroptosis [60], selective induction of cuproptosis may also be adopted as a potential treatment strategy for CRC.

However, there are some limitations to our study. Firstly, we only explored the potential cuproptosis regulation function of TIGD1 in CRC without further investigation of the other six prognostic cuproptosis-related genes. Secondly, our study was limited to in vitro experiments and did not include any in vivo experiments.

## 5. Conclusions

Our study investigated the expression patterns of ten cuproptosis markers and developed a robust prognostic predictive model using seven PCRGs. Additionally, we conducted a preliminary analysis on the regulatory role of TIGD1 in cuproptosis. The signature proposed in this study has potential clinical applications as a useful biomarker and candidate target for colorectal cancer. It can assist in predicting the immunotherapy sensitivity of CRC patients in clinical practice. Importantly, targeted inhibition of TIGD1 to enhance the sensitivity of cuproptosis in CRC cells may also be a promising strategy for treating CRC.

## Figures and Tables

**Figure 1 cancers-15-02286-f001:**
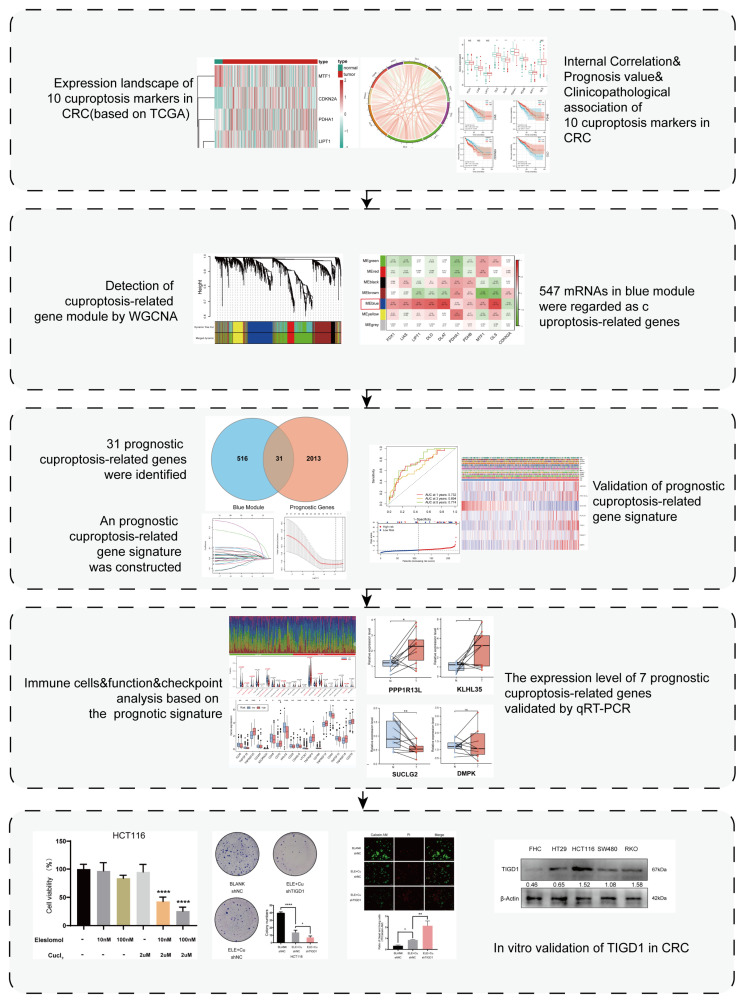
Study flowchart. * *p* < 0.05, ** *p* < 0.01, *** *p* < 0.001 and **** *p* < 0.0001; ns, no significance.

**Figure 2 cancers-15-02286-f002:**
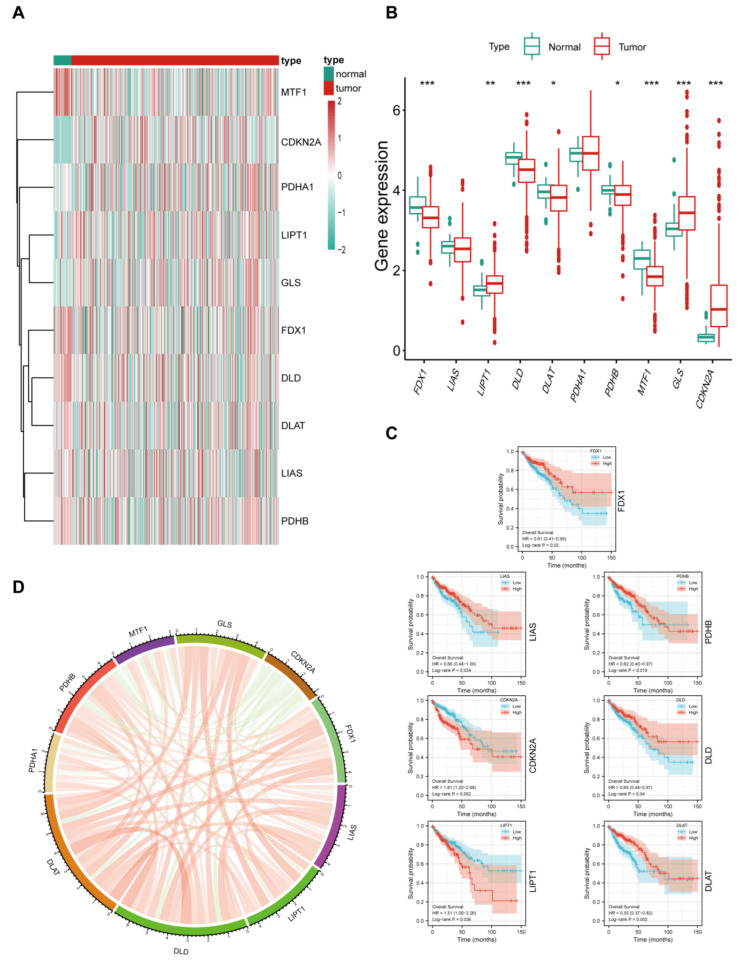
Expression landscapes of cuproptosis markers in CRC; (**A**) expression levels of ten cuproptosis CRC markers; (**B**) histogram of the ten differentially expressed cuproptosis markers in CRC; (**C**) Kaplan–Meier survival curves depict the OS rates of patients with CRC with high (red line) and low (blue line) expression levels of DFX1, LIAS, PDHB, CDKN2A, DLD, LIPT1, and DLAT; (**D**) chord graph depicting the Spearman correlation analysis of the ten cuproptosis CRC markers. * *p* < 0.05, ** *p* < 0.01, and *** *p* < 0.001.

**Figure 3 cancers-15-02286-f003:**
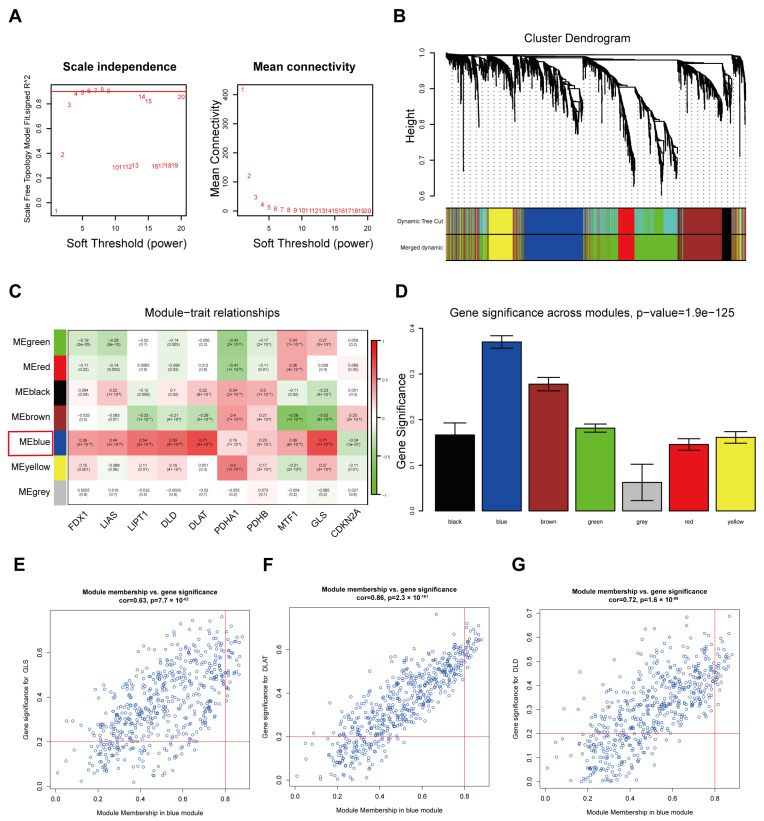
Identification of cuproptosis-related genes of CRC in the TCGA dataset using WGCNA; (**A**) analysis of the scale-free fit index for various soft-thresholding powers (left) and analysis of the mean connectivity for various soft-thresholding powers (right); (**B**) dendrogram of all DEGs clustered based on dissimilarity measures; (**C**) heatmap of the correlation between module eigengenes and ten cuproptosis markers; (**D**) correlation analysis between modules and cuproptosis markers; (**E**–**G**) scatter plots of module eigengenes in the selected modules (each dot represents a gene).

**Figure 4 cancers-15-02286-f004:**
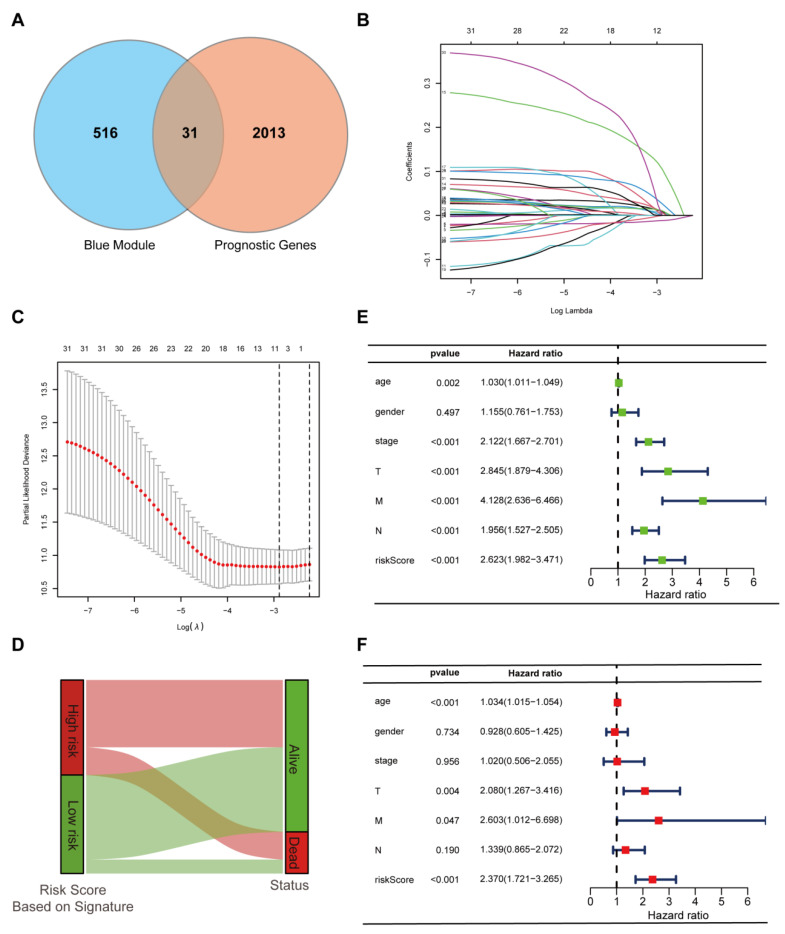
Construction of a 7-cuproptosis-related gene signature and the analysis of its independent prognostic potential; (**A**) Venn diagram identifying the common CRC prognostic genes and cuproptosis-related genes of CRC; (**B**) Cvfit curves showing LASSO coefficient profiles of the 31 candidate genes; (**C**) Selection of the optimal LASSO parameter lambda; vertical lines indicate the optimal values. (**D**) Sankey diagram representing the detailed connection between clusters, risk groups, and the live status of patients with CRC; (**E**,**F**) results of the univariate and multivariate Cox regression analyses with regard to the OS of the 7-cuproptosis-related gene signature. The red and green boxes stand for their harzard ratio.

**Figure 5 cancers-15-02286-f005:**
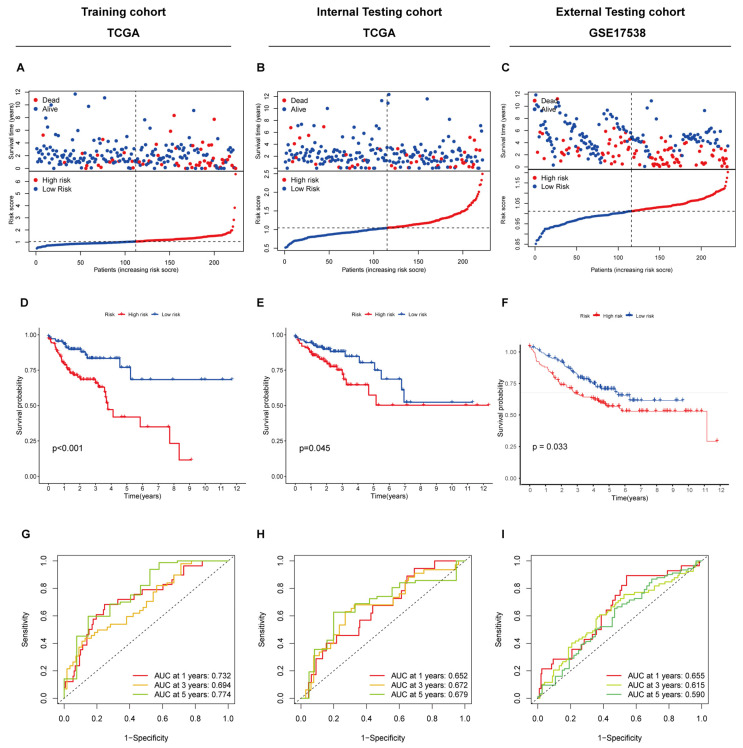
Construction and validation of the cuproptosis-related gene signature in the training and testing groups; (**A**–**C**) Distribution of risk scores and OS rates in the training and testing groups; (**D**–**F**) Kaplan–Meier curves for survival status and survival time in the training and testing groups; (**G**–**I**) the ROC curve shows the potential of the prognostic cuproptosis-related gene signature in predicting 1-, 2-, and 3-year OS in the training and testing groups.

**Figure 6 cancers-15-02286-f006:**
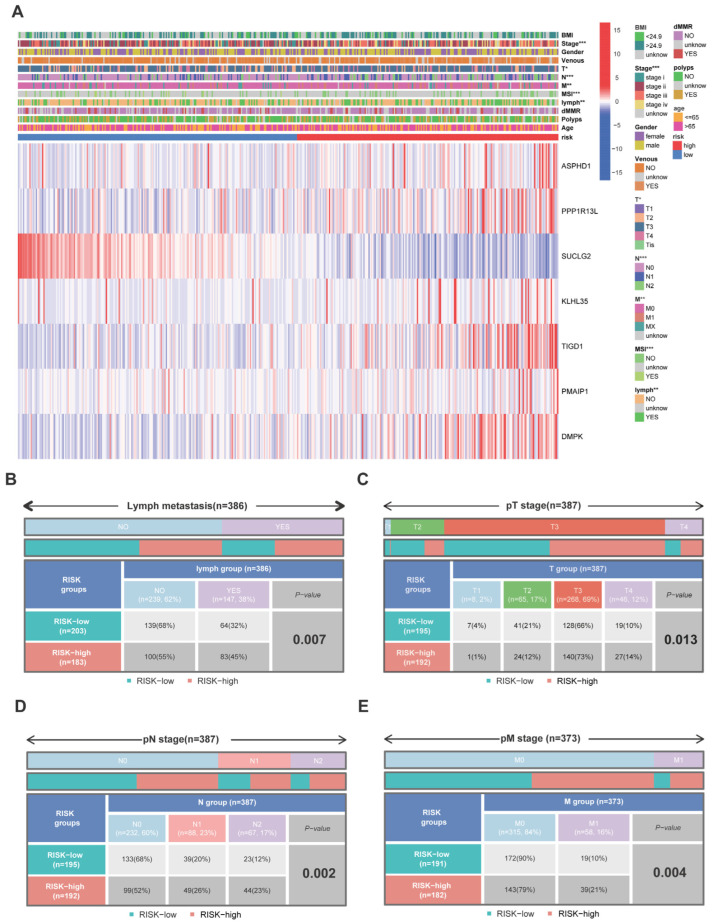
Correlation analysis between the prognostic signature and different clinicopathological characteristics in the TCGA cohort; (**A**) a heatmap of the distribution of 12 different clinicopathological characteristics of the risk groups of each patient based on the signature; (**B**–**E**) a rectangle diagram depicting the significant differences between the risk scores in patients with CRC stratified by lymph invasion, T stage, N stage, and M stage; * *p* < 0.05, ** *p* < 0.01, and *** *p* < 0.001.

**Figure 7 cancers-15-02286-f007:**
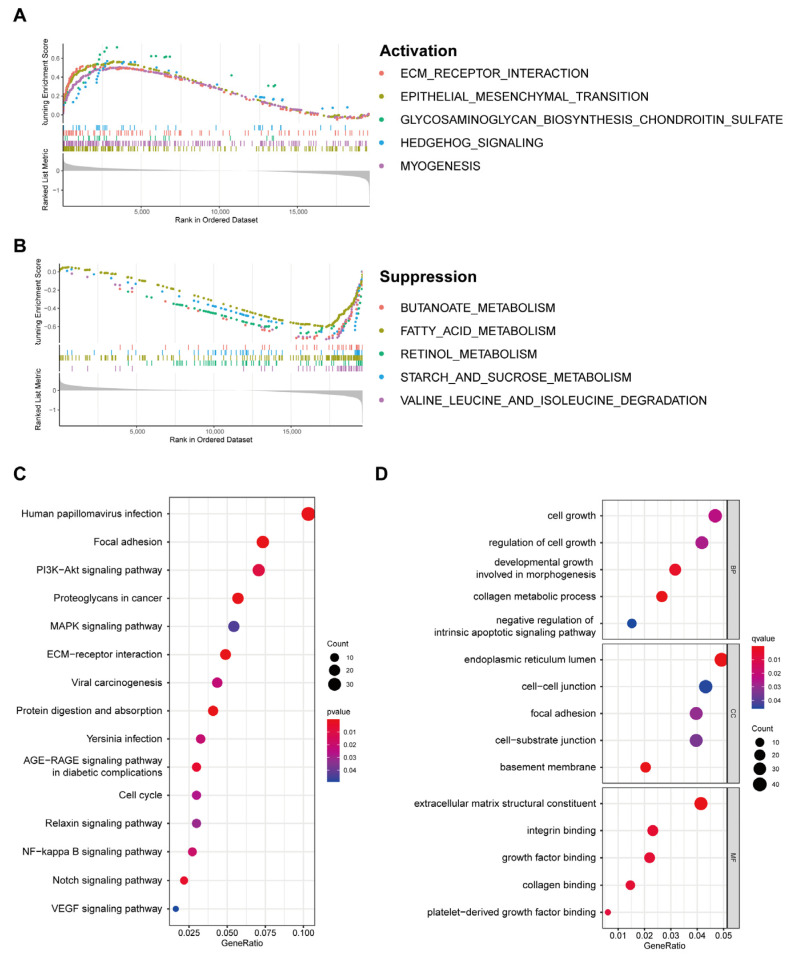
Biological, functional, and pathway enrichment analysis of the high-risk and low-risk groups based on the cuproptosis-related gene signature; (**A**) GSEA revealed significant enrichment of pathways activated in high-risk patients with CRC; (**B**) GSEA showing significant number of pathways suppressed in high-risk patients with CRC; (**C**) KEGG analysis revealing that many cancer progression pathways were enriched; (**D**) GO analysis showing the enrichment of many cancer-growth-related biological processes and molecular functions.

**Figure 8 cancers-15-02286-f008:**
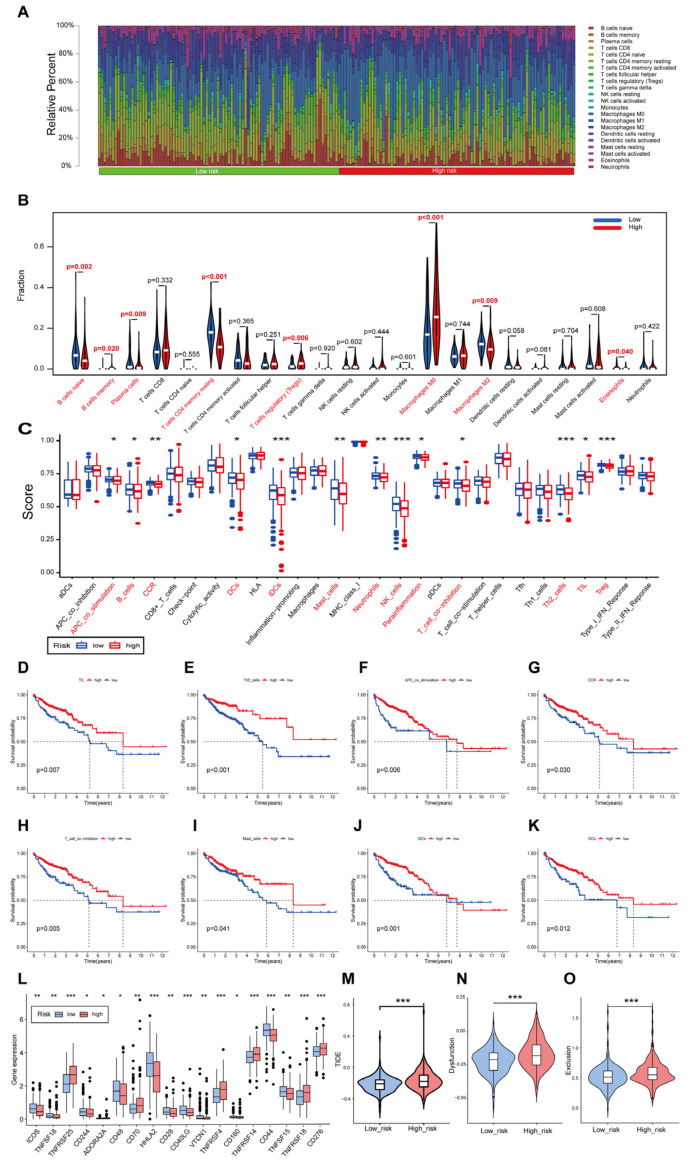
Analysis of immune-related landscapes in patients with CRC based on the cuproptosis-related gene signature; (**A**) bar graphs exhibiting the distribution of tumor-infiltrating immune cells between the high-risk and low-risk groups based on the cuproptosis-related gene signature; (**B**) Vioplot for comparing the 22 immune cells between the high-risk and low-risk groups of patients with CRC; (**C**) boxplots comparing the diverse immune cell subpopulations, related pathways, and functions between the high-risk and low-risk groups of patients with CRC; (**D**–**K**) Kaplan–Meier survival curves showing the OS rates of patients with CRC according to the differences of diverse immune cell subpopulations, related pathways, and functions; (**L**) boxplots for comparing immune checkpoint genes between the high-risk and low-risk groups; (**M**–**O**) Vioplot of TIDE results between the high-risk and low-risk groups of patients with CRC.* *p* < 0.05, ** *p* < 0.01, and *** *p* < 0.001.

**Figure 9 cancers-15-02286-f009:**
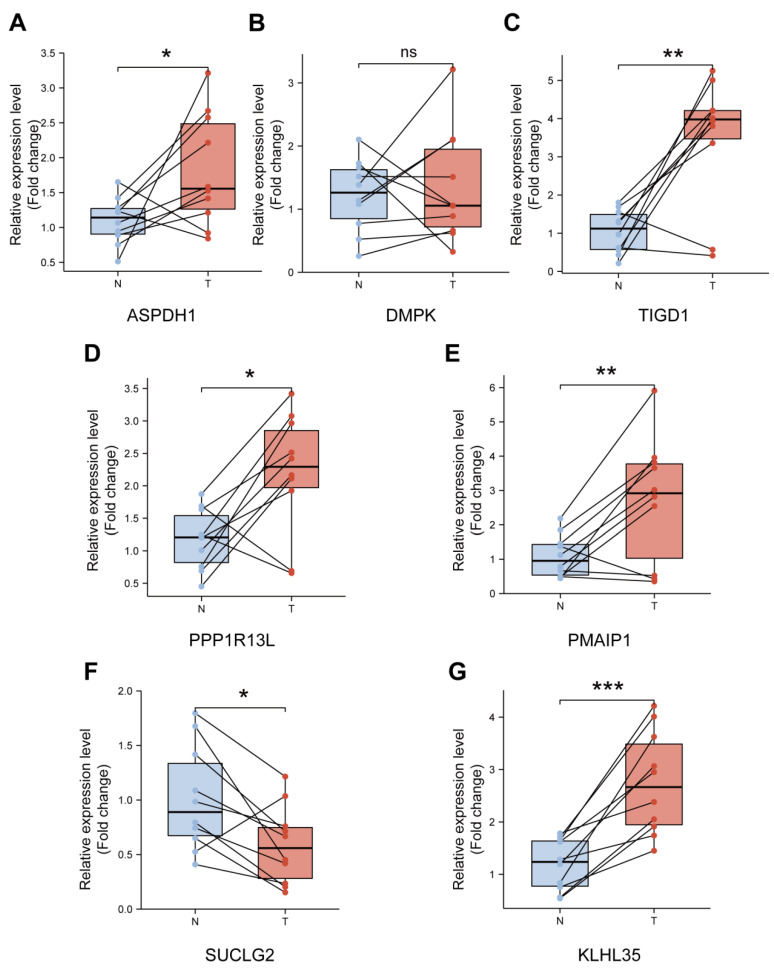
Validation of the expression levels of the seven PCRGs in patient samples; (**A**–**G**) qRT-PCR showing the expression of ASPDH1, DMPK, TIGD1, PPP1R13L, PMAIP1, SUCLG2, and KLHL35 in ten pairs of CRC tissue samples and adjacent normal tissues; (Blue box: pericardial tissues; red box: tumor tissue); * *p* < 0.05, ** *p* < 0.01, *** *p* < 0.001; ns, no significance.

**Figure 10 cancers-15-02286-f010:**
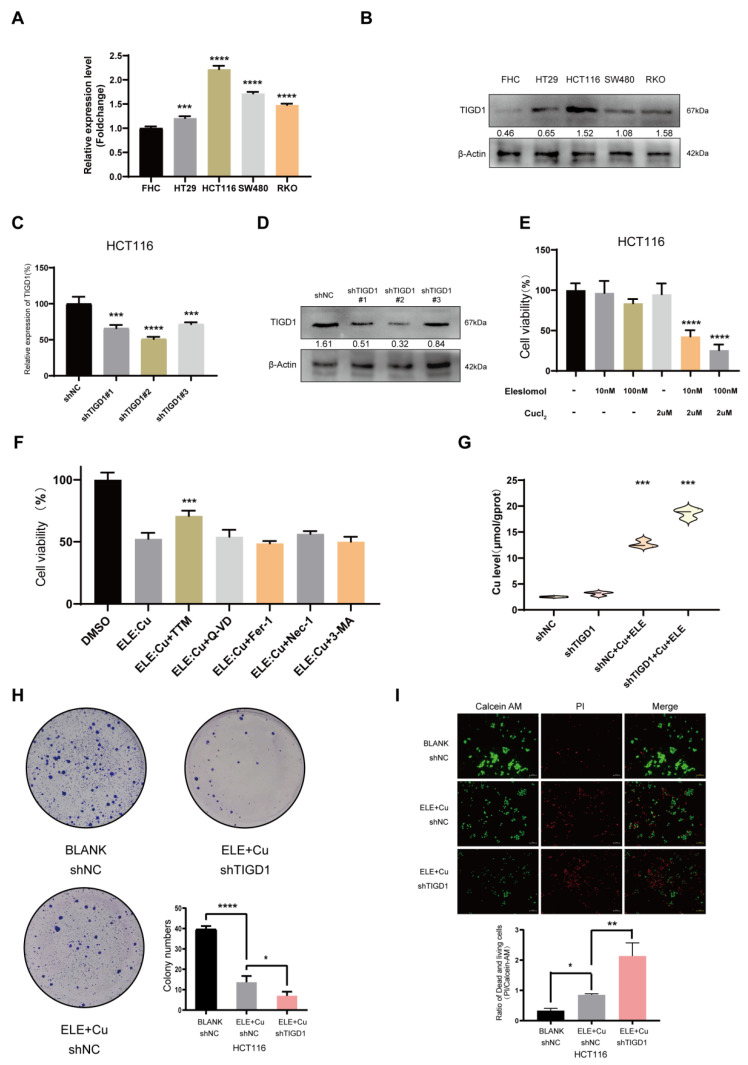
Verification of elesclomol-induced cuproptosis and the identification of TIGD1 in regulating cuproptosis in CRC; (**A**) qRT-PCR showing the mRNA expression of TIGD1 in human normal colon epithelial cells and four human CRC cell lines (HT29, HCT116, SW480, and RKO); (**B**) representative western blotting of TIGD1 protein in human normal colon epithelial cells and four human CRC cell lines (HT29, HCT116, SW480, and RKO),uncropped western blotting in Appendix A; (**C**) qRT-PCR showing the significant knockdown of TIGD1 in HCT116 after the transfection of short hairpin RNA; (**D**) representative western blotting of TIGD1 protein in HCT116 after the transfection of short hairpin RNA, uncropped western blotting in Appendix A; (**E**) cell viability after exposure of HCT116 cells to different concentrations of elesclomal (10 nM, 100 nM) and Cucl_2_ (2 µM), measured using the CCK-8 assay; (**F**) cell viability of HCT116 cells after the treatment with DMSO, tetrathiomolybdate, Q-VD-OPH, necrostatin-1, ferrostatin-1, and 3-methyladenine, then were exposure to 10 nM elesclomal and 2 µM Cucl_2_; (**G**) concentration of copper on treatment with 10 nM elesclomol and 2 µM CuCl_2_ in control/shTIGD1 groups; (**H**) representative images for evaluating the colony formation capacity of CRC cells using colony formation assays. Colony numbers in each group were analyzed using by ImageJ 1.51; (**I**) the ratio of dead/living HCT116 cells (fluorescence excitation by calcein–AM/PI) indicated the viability of control/shTIGD1 groups after treatment with 10 nM elesclomol and 2 µM CuCl_2,_ Green fluorescence represents live cells and red fluorescence represents dead cells; * *p* < 0.05, ** *p* < 0.01, *** *p* < 0.001, **** *p* < 0.0001.

**Table 1 cancers-15-02286-t001:** Clinical characteristics of patients with CRC in the training and test cohorts.

Characteristics	Training Group		Testing Group		*p*-Value
	No.	%	No.	%	
Age	—		—		—
≤65	93		83		>0.05
>65	119		129		—
Gender	—		—		—
Male	121		107		>0.05
Female	91		105		—
AJCC Stage	—		—		—
I	36		37		>0.05
II	83		82		—
III	53		64		—
IV	35		23		—
T stage	—		—		—
T1	4		6		>0.05
T2	38		36		—
T3	141		149		—
T4	29		20		—
N stage	—		—		—
N0	127		126		>0.05
N1	44		53		—
N2	41		33		—
M stage	—		—		—
M0	155		160		>0.05
M1	35		23		—

## Data Availability

All original contributions presented in this study are included in the manuscript and Appendix A. Further inquiries can be directed to the corresponding authors.

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
