# Peer review of "TIGD1 Function as a Potential Cuproptosis Regulator Following a Novel Cuproptosis-Related Gene Risk Signature in Colorectal Cancer"

_cancers, 2023, doi:10.3390/cancers15082286_

Round 1
Reviewer 1 Report
A study by Wu et a; investigates the role of TIGD1 as a cuproptosis regulator following a novel cuproptosis-related gene risk signature in colorectal cancer. As cuproptosis is a novel pathway of cell death, without extensive experimental validation yet, the study approach is novel and interesting. The bioinformatic part is well performed, although it predominantly bases on previous discoveries and analyses.
Specific comments:
1. In Materials and Methods: "Appropriate inhibitors were added after plating for 12 h." Was it prior to treatment with elesclomol? It should be clarified.
2. Fig. 10 - "knockdown efficiency" sounds awkward. Does it mean that control is knockdowned with full efficiency?
3. Fig. 10I needs quantification.
4. Please rpovide quantification of all Western blots.
5. Fig. 10D - please adjust upper labels for each lane.
Author Response
We attached figures in WORD profile.
Dear editor and reviewers.
Please find attached revised manuscript entitled “TIGD1 Function as a Potential Cuproptosis Regulator Following a Novel Cuproptosis-Related Gene Risk Signature in Colorectal Cancer”. We are grateful for the referees’ helpful suggestions and have revised this manuscript accordingly. We believe that this significantly strengthened manuscript is now suitable for publication in Cancers. We marked the all the changes using the “Track Changes” function in the revised manuscript with marks. And the point by point response to all reviewers are listed below.
Thank you very much for your consideration.
Sincerely yours,
Dr. Liying Song
The Third Xiangya Hospital of Central South University
Point by point response:
Reviewer1:
- In Materials and Methods: "Appropriate inhibitors were added after plating for 12 h." Was it prior to treatment with elesclomol? It should be clarified.
Response: We apologize for any potential misunderstanding about the methods of Chemical Reagent Assay. The relative description have been modified to make it delivered more clearly.In brief, we planted the cells and incubated for 24 h.And the inhibitors were added and incubated for 12 h. Subsequently, elesclomol and cucl2 were added.
The relevant description has been added in lines 260-268.
- 10 - "knockdown efficiency" sounds awkward. Does it mean that control is knockdowned with full efficiency?
Response: Since "knockdown efficiency" sounds awkward. We have replaced it with “Relative expression of TIGD1(%)” in Fig.10C. Which could be more direct and clear. In addition,control is the CRC cells transfected with negative control shRNA.
Figure 10B
- 10I needs quantification.
Response: Thanks for your valuable suggestion. We have quantified Fig.10I as request.
Figure 10I
- Please rpovide quantification of all Western blots.
Response: Thanks for your helpful suggestion. All western blot panels(Fig.10B and Fig.10D) have been quantified.
Figure 10B
Figure 10D
- Fig. 10D - please adjust upper labels for each lane.
Response: Thanks for your helpful suggestion. We have adjust upper labels of each lane in Fig.10D accordingly.
Figure 10D
Thank you very much for your consideration. I look forward to hearing from you.
Sincerely,
Dr. Liying Song
The Third Xiangya Hospital of Central South University

Reviewer 2 Report
The manuscript was interesting to read and have a new approach. However, some comments for the attention of the authors are listed below on each part of the manuscript:
Simple Summary: This should be written in one paragraph before the Abstract in layman’s terms to explain why the research is being suggested, what the authors aim to achieve, and how the findings from this research may impact the research community.
Please do not cite references in the simple summary
Abstract:
1. Authors do not provide a clear explanation of why understanding the role of cuproptosis in CRC is important or how their findings may impact the research community.
2. The abstract could benefit from more specific details about the methodology used in the study, such as the sample size and the statistical methods used to identify the prognostic genes.
3. The authors could provide more context about the current state of knowledge in the field of cuproptosis and CRC. Finally, the abstract could be improved by including more specific information about the implications of the findings for clinical practice or future research.
Introduction: there are some areas where the introduction can be improved.
1. while the authors provide a good overview of the existing literature on cell death pathways and their associations with CRC, they could have cited more recent studies in the field to strengthen their arguments.
2. the introduction could benefit from a clearer statement of the research question or hypothesis that the study seeks to address.
3. The introduction also lacks a clear justification for the use of the two analytical methods, WGCNA and LASSO-Cox regression, in identifying novel genes in cuproptosis regulation and establishing a prognostic signature. The authors briefly mention these methods without explaining why they are appropriate or necessary for the study.
4. the introduction could benefit from a more detailed explanation of the potential clinical applications of the study findings. The authors briefly mention that their study provides novel insights for CRC diagnosis and treatment, but they do not elaborate on how the identification of novel genes in cuproptosis regulation and the establishment of a prognostic signature can translate into clinical benefits for patients.
5. Data Acquisition: While it is clear that the RNA-seq data of 512 CRC samples were downloaded from the TCGA website, it would be helpful to know the date of the data download and any quality control measures taken before proceeding with the analysis.
6. Differential Expression Analysis: The criteria used for identifying DEmRNAs seem to be standard; however, there is no mention of any adjustments made for multiple testing. It would be useful to know whether any such adjustments were made and, if so, what method was used.
7. Gene Set Variation Analysis (GSVA): The authors have provided information about the R package and databases used for GSVA; however, there is no explanation of what GSVA is or why it was used in this study. It would be helpful to briefly introduce GSVA and provide a rationale for using it in this context.
8. Construction of the Cuproptosis-Related Prognostic Signature: The section on the construction of the prognostic signature is quite detailed, but it would be helpful to know how the candidate genes were selected for Cox univariate analysis.
9. Additionally, it would be useful to know what statistical method was used to compare the OS rates between the risk groups in the testing cohort.
10. Functional Enrichment Analysis: The authors have provided information about the R package used for functional enrichment analysis, but there is no explanation of what functional enrichment analysis is or how it was used in this study. It would be helpful to briefly introduce functional enrichment analysis and provide a rationale for using it in this context.
Results:
1. the paper should provide more information on the ten markers selected for analysis, as well as their biological relevance in CRC. This would help readers understand the rationale behind selecting these markers and how they contribute to the pathogenesis of CRC.
2. the authors could provide a more detailed discussion of the differential expression patterns of the eight genes and how they relate to the pathogenesis of CRC.
3. the paper could benefit from a more detailed explanation of the WGCNA method used to identify novel proptosis-related mRNAs in CRC. This would help readers understand the strengths and limitations of this approach, as well as the relevance of the identified mRNAs to the pathogenesis of CRC.
4. the paper could benefit from a more detailed discussion of the limitations of the study, as well as the implications of the findings for clinical practice. Specifically, the authors could discuss how the identified cuproptosis markers could be used for early detection, diagnosis, and prognosis of CRC, as well as the potential for developing novel therapeutic targets.
5. the authors should provide more context and discussion of the available literature on the topic, including previous studies on the expression landscape of cuproptosis markers in CRC, as well as their relevance to clinical outcomes. This would help readers understand the novelty and contribution of the current study to the field.
6. the authors state that they used ssGSEA to compare the correlation between risk groups and immune statuses. However, they do not provide sufficient detail about the statistical analysis and how the ssGSEA method was applied. The reader is left with many questions about the methodology, including how the authors determined which components of antigen presentation were significantly different between low- and high-risk CRC patients.
7. the authors only briefly mention the immune cells that were significantly different between the two groups without providing more information on which cells were different and why this finding is important.
8. The authors suggest that a lower level of immune status may result in worse survival, but they do not provide sufficient evidence to support this claim. They only present a correlation between immune statuses and risk groups and do not provide survival data for the low- and high-risk groups. Additionally, the authors state that the immune ability was relatively inhibited in the high-risk group, but they do not provide any evidence to support this claim.
9. Regarding Figures 9 and 10, the authors provide clear explanations of the experimental design and the results obtained. However, the presentation of the data could be improved. For example, the figures are difficult to read, and the legends could be more informative.
Conclusions:
1. The paper does not clearly state the research question that the study aims to address. It is important to have a clear research question that guides the study's design, methodology, and analysis to ensure that the study is focused and rigorous.
2. The paper briefly mentions the identification of cuproptosis markers and genes, as well as the construction of a prognostic predictive model. However, there is no explanation of the methods used to identify these markers and genes, or how the predictive model was constructed. This lack of detail makes it difficult for readers to assess the validity and reliability of the findings.
3. The conclusion only briefly summarizes the findings of the study, without providing any in-depth analysis or discussion of the results. For example, the paper mentions the relationship between the risk signature and immune landscapes but does not elaborate on this relationship or its implications for clinical practice.
4. The paper does not provide a sufficient comparison of its findings with existing literature. This makes it difficult for readers to assess the novelty and significance of the study's contributions.
5. While the paper claims that the findings may be useful for predicting outcomes and treatment sensitivity in clinical practice, there is no discussion of how this might be achieved or what implications the findings have for patient care.
Author Response
Dear editor and reviewers.
Please find attached revised manuscript entitled “TIGD1 Function as a Potential Cuproptosis Regulator Following a Novel Cuproptosis-Related Gene Risk Signature in Colorectal Cancer”. We are grateful for the referees’ helpful suggestions and have revised this manuscript accordingly. We believe that this significantly strengthened manuscript is now suitable for publication in Cancers. We marked the all the changes using the “Track Changes” function in the revised manuscript with marks. And the point by point response to all reviewers are listed below.
Thank you very much for your consideration.
Sincerely yours,
Dr. Liying Song
The Third Xiangya Hospital of Central South University
Point by point response:
Reviewer2:
Simple Summary: This should be written in one paragraph before the Abstract in layman’s terms to explain why the research is being suggested, what the authors aim to achieve, and how the findings from this research may impact the research community.
Response: Thanks for your valuable suggestion.We have added simple summary in Lines 26-33 as request.
- Authors do not provide a clear explanation of why understanding the role of cuproptosis in CRC is important or how their findings may impact the research community.
Response: Due to the words limitation of abstract,we have revised abstract through adding the short description: There is emerging evidence indicating that cuproptosis has a significant regulatory function in the onset and progression of cancer. However, it is still unclear how cuproptosis regulates cancer and whether other genes are involved in the regulation.
The relevant description has been added in lines 37-40.
- The abstract could benefit from more specific details about the methodology used in the study, such as the sample size and the statistical methods used to identify the prognostic genes.
Response: We have revised accordingly, We added some descriptions like: Using the TCGA-COAD dataset of 512 samples,Using least absolute shrinkage and selection operator (LASSO)–Cox regression analysis.Using Kaplan–Meier survival analysis. We believe these details about the methodology could help to improve the quality of our abstract.
The relevant description has been added in lines 40-45.
- The authors could provide more context about the current state of knowledge in the field of cuproptosis and CRC. Finally, the abstract could be improved by including more specific information about the implications of the findings for clinical practice or future research.
Response: We have revised accordingly. At the end of abstract, we added: Since certain concentration of copper in CRC cells is important, cuproptosis may provide a new target for cancer therapy. And this study may provide novel insights in the treatment of CRC.
The relevant description has been added in lines 55-58.
Introduction: there are some areas where the introduction can be improved.
- while the authors provide a good overview of the existing literature on cell death pathways and their associations with CRC, they could have cited more recent studies in the field to strengthen their arguments.
Response: We have cited more recent studies about the cell death pathways and their associations with CRC. Just like ref.8, ref.9,ref.11, and reference 13.(All published after 2021).
The relevant description has been added in lines 72-77.
- the introduction could benefit from a clearer statement of the research question or hypothesis that the study seeks to address.
Response: We have represented a clearer hypothesis: Therefore, we have reason to hypothesize that there must be more novel genes involved in the process of cuproptosis in CRC. And understand how they participate in the regulation of cuproptosis in CRC is unquestionably important.
The relevant description has been added in lines 72-77.
- The introduction also lacks a clear justification for the use of the two analytical methods, WGCNA and LASSO-Cox regression, in identifying novel genes in cuproptosis regulation and establishing a prognostic signature. The authors briefly mention these methods without explaining why they are appropriate or necessary for the study.
Response: In revised manuscript, i have justified in discussion that why should WGCNA and and LASSO-Cox regression were conducted. As showed in lines 587- 593. Firstly, WGCNA is an efficient gene screening method that utilizes transcriptome expression matrices. And using WGCNA, we identified the core module with 537 mRNAs that closely related to 10 makers of cuproptosis from more than 20000 mRNA. But 537 mRNAs are still too much for conducting subsequent validation. Base on this point , we conducted univariate COX analysis, LASSO-Cox regression to narrow down the list. And finally identified 7 cuproptosis-related genes.
The relevant description has been added in lines 587- 593.
- the introduction could benefit from a more detailed explanation of thepotential clinical applications of the study findings. The authors briefly mention that their study provides novel insights for CRC diagnosis and treatment, but they do not elaborate on how the identification of novel genes in cuproptosis regulation and the establishment of a prognostic signature can translate into clinical benefits for patients.
Response: At the end of the introduction, we added some descriptions to illustrate how this study might benefits CRC patients: The establishment of this signature and the identification of TIGD1 in cuproptosis regulation may ultimately translate into better clinical guidance for CRC patients. By providing a more accurate prognosis and guiding treatment selection, this study has the potential to enhance treatment sensitivity and improve the survival rates of CRC patients.
The relevant description has been added in lines 112-116.
- Data Acquisition: While it is clear that the RNA-seq data of 512 CRC samples were downloaded from the TCGA website, it would be helpful to know the date of the data download and any quality control measures taken before proceeding with the analysis.
Response: I have added the data of data download and introduce how we normalized the data as quality control.
The relevant description has been added in lines 122-130.
- Differential Expression Analysis: The criteria used for identifying DEmRNAs seem to be standard; however, there is no mention of any adjustments made for multiple testing. It would be useful to know whether any such adjustments were made and, if so, what method was used.
Response: In this study, we applied differential expression analyses using “limma” package. And we used false discovery rate (FDR) correction for adjustments. The criteria for DEmRNAs were |log 2(fold change)| > 1 and FDR < 0.05.We have revised the method section of Differential Expression Analysis.
The relevant description has been added in lines 133-139.
- Gene Set Variation Analysis (GSVA): The authors have provided information about the R package and databases used for GSVA; however, there is no explanation of what GSVA is or why it was used in this study. It would be helpful to briefly introduce GSVA and provide a rationale for using it in this context.
Response: We apologize for any confusion about the GSVA. We conducted massive bioinformatic analyses to identify the full landscape of cuproptosis. However, we deleted some of the data since many of them are similar with others, including GSVA. Therefore, in revised manuscript, we have also deleted the description of Gene Set Variation Analysis in Materials and Methods.
- Construction of the Cuproptosis-Related Prognostic Signature: The section on the construction of the prognostic signature is quite detailed, but it would be helpful to know how the candidate genes were selected for Cox univariate analysis.
Response: Using WGCNA, we identified the core module with 537 mRNAs that closely related to 10 makers of cuproptosis from more than 20000 mRNA. Then we chose this 537 mRNAS as candidate genes for subsequent cox univariate analysis.
- Additionally, it would be useful to know what statistical method was used to compare the OS rates between the risk groups in the testing cohort.
Response: We applied log-rank test to compare the OS rates between the risk groups in the testing cohort. Here is the introduction of log-rank test: Log-rank test, which is a nonparametric test that compares the survival distributions of two or more groups.It assumes that the hazard ratios are constant over time and that the survival curves do not cross. The test statistic is based on the difference between the observed and expected number of events in each group, and it follows a chi-squared distribution with one degree of freedom.
The relevant description has been added in lines 176-180.
- Functional Enrichment Analysis: The authors have provided information about the R package used for functional enrichment analysis, but there is no explanation of what functional enrichment analysis is or how it was used in this study. It would be helpful to briefly introduce functional enrichment analysis and provide a rationale for using it in this context.
Response: We have revised the Functional Enrichment Analysis in Materials and Methods. In revised manuscripts, we explained functional enrichment analysis is or how it was used in this study: KEGG is a database that helps researchers understand high-level biological functions and contains pathway maps and associated metabolic and signaling pathways. GO is a standardized vocabulary that describes the attributes of genes and gene products across all organisms in terms of their biological processes (BP), cellular component (CC), and molecular functions (MF). It provides a basic guidance for subsequent functional analysis.During functional enrichment analysis, differentially expressed genes were selected and compared to the terms in either the Gene Ontology or KEGG database. The number of genes that matched each term was determined, and hypergeometric tests were used to identify significantly enriched entries. An adjusted p-value threshold of less than 0.05 was used to determine statistical significance.
The relevant description has been added in lines 198-210.
Results:
- the paper should provide more information on the ten markers selected for analysis, as well as their biological relevance in CRC. This would help readers understand the rationale behind selecting these markers and how they contribute to the pathogenesis of CRC.
Response: We added more information and citations on four markers(DFX1,CDKNA2,PDHA and GLS) as well as their biological relevance in CRC at Discussion section. And there is no literature ever reported the other six genes in CRC.
The relevant description has been added in lines 566-579.
- the authors could provide a more detailed discussion of the differential expression patterns of the eight genes and how they relate to the pathogenesis of CRC.
Response: We added a more detailed discussion of the differential expression patterns of the eight genes and how they might be relate to the pathogenesis of CRC:More importantly, we discovered that FDX1, DLD, and MTF1 were markedly downregulated in CRC. This leads us to infer that their function in CRC may be related to inhibiting cancer progression. In contrast, GLS and CDKN2A showed significantly upregulated expression levels in CRC patients. This suggests that these two genes could serve as potential clinical diagnostic markers for CRC. Furthermore, they might increase the malignancy of CRC by promoting its tumorigenesis and development.
The relevant description has been added in lines 552-558.
- the paper could benefit from a more detailed explanation of the WGCNA method used to identify novel proptosis-related mRNAs in CRC. This would help readers understand the strengths and limitations of this approach, as well as the relevance of the identified mRNAs to the pathogenesis of CRC.
Response: We used WGCNA to identify novel cuproptosis-related mRNAs in CRC. And we identified 7 modules and found that blue module were highly correlated with 10 cuproptosis markers. And there are 537 mRNAs in blue module. So we believed that these 537 mRNAs have the similar expression pattern with each other and may connected with cuproptosis. We have introduced the detail of WGCNA in Materials and Methods, Results, and the Discussion.
The relevant description can be seen in lines 147-160, lines 349-361 and lines586-592.
- the paper could benefit from a more detailed discussion of the limitations of the study, as well as the implications of the findings for clinical practice. Specifically, the authors could discuss how the identified cuproptosis markers could be used for early detection, diagnosis, and prognosis of CRC, as well as the potential for developing novel therapeutic targets.
Response: The discussion of the limitation could truly improve the reliability and validity of our study. And we have added it at the end of Discussions:However, there are some limitations about our study. Firstly, we only explored the potential cuproptosis regulation function of TIGD1 in CRC, without further investigation of the other six prognostic cuproptosis-related genes. Secondly, our study was limited to in vitro experiments and did not include any in vivo experiments.
The relevant description has been added in lines 552-558.
Meanwhile, we also discussed about how it could be used as potential clinical applications of CRC:For instance, these genes may serve as biomarkers for detecting CRC at an early stage. And just like the clinical application targeting ferroptosis[60], selectively induction of cuproptosis may also be adopted as a potential treatment strategy in CRC.
The relevant description has been added in lines 655-658.
- the authors should provide more context and discussion of the available literature on the topic, including previous studies on the expression landscape of cuproptosis markers in CRC, as well as their relevance to clinical outcomes. This would help readers understand the novelty and contribution of the current study to the field.
Response: We have included three previous studies of cuproptosis in CRC. And we also discussed their limitations. We believe this could help readers understand the novelty and strength of our study.
The relevant description has been added in lines 640-645.
- the authors state that they used ssGSEA to compare the correlation between risk groups and immune statuses. However, they do not provide sufficient detail about the statistical analysis and how the ssGSEA method was applied. The reader is left with many questions about the methodology, including how the authors determined which components of antigen presentation were significantly different between low- and high-risk CRC patients.
Response: The relevant statement of how ssGSEA were applied were added in Materials and Methods:GSEA (ssGSEA) was conducted using the 'GSVA' R package based on a previous study[23] to analyze the infiltrating scores of 16 immune cells and the activities of 13 immune-related pathways between the two risk groups. And the Mann- Whitney test using the BH method adjusted p-value was adopted to measure the ssGSEA scores.
The relevant description has been added in lines 191-193 and lines 307-308.
- the authors only briefly mention the immune cells that were significantly different between the two groups without providing more information on which cells were different and why this finding is important.
Response: In the Results and Discussion section of this study, we mentioned that there are a total of eight types of immune cells that show differential expression among different risk groups, including naive/memory B cells, plasma cells, CD4 memory/regulatory T cells, M0/M2 macrophages, and eosinophils. Additionally, we also cited three studies (Ref. 50, Ref. 51, Ref. 52) in the discussion to demonstrate the importance of immune cell expression in cancers.
- The authors suggest that a lower level of immune status may result in worse survival, but they do not provide sufficient evidence to support this claim. They only present a correlation between immune statuses and risk groups and do not provide survival data for the low- and high-risk groups. Additionally, the authors state that the immune ability was relatively inhibited in the high-risk group, but they do not provide any evidence to support this claim
Response: In Figures 5D, E, and F, we noticed that through Kaplan–Meier analysis, high-risk groups had short overall survival times in the training, validation, and external validation cohorts. Therefore, We claimed that patients in the high-risk group had worse survival rate. Additionally, in Figure 8B, we found a significant decrease in the abundance of immune cells among patients in the high-risk group. In Figures 8D-8K, we grouped CRC patients based on the high or low expression of these immune cells and found that patients with lower expression levels of these immune cells typically had poor survival outcomes. Since the quantity of many immune cells is positively correlated with the body's immune status(the adoptive cellular immunotherapy we used in clinical perfectly supports this point of view). Therefore, based on these results, we have reason to believe that patients from high-risk group with a lower level of immune status may result in worse survival.
- Regarding Figures 9 and 10, the authors provide clear explanations of the experimental design and the results obtained. However, the presentation of the data could be improved. For example, the figures are difficult to read, and the legends could be more informative.
Response: We have revised Figure 9 and Figure 10 as request: In figure 9, we have enhanced the fonts and slightly changed the position of each figures.In figure 10B and Figure 10D, we provided the quantification of all Western blots. In figure 10G, we added statistically difference. In figure 10I, we also quantified all figures.
Meanwhile, we re-edited the legends of Figure 9 and Figure 10.
The relevant description has been changed in lines 937-941 and lines 943-964.
Conclusions:
- The paper does not clearly state the research question that the study aims to address. It is important to have a clear research question that guides the study's design, methodology, and analysis to ensure that the study is focused and rigorous.
Response: We have state the question and the study we aimed to address in introduction: we have reason to hypothesize that there must be more novel genes involved in the process of cuproptosis in CRC. And understand how they participate in the regulation of cuproptosis in CRC is unquestionably important.
The relevant description has been added in lines 72-77.
- The paper briefly mentions the identification of cuproptosis markers and genes, as well as the construction of a prognostic predictive model. However, there is no explanation of the methods used to identify these markers and genes, or how the predictive model was constructed. This lack of detail makes it difficult for readers to assess the validity and reliability of the findings.
Response: We identified novel cuproptosis-related genes using WGCNA, Cox univariate analysis and LASSO–Cox regression analysis. In the Materials and Methods section, we provide a detailed description of the methodology about these analyses, with the aim of enhancing the reader's understanding of our gene screening and model constructing process.
- The conclusion only briefly summarizes the findings of the study, without providing any in-depth analysis or discussion of the results. For example, the paper mentions the relationship between the risk signature and immune landscapes but does not elaborate on this relationship or its implications for clinical practice.
Response: In this study, using Tumor Immune Dysfunction and Exclusion (TIDE) analysis, we showed that high-risk patients with CRC have a higher potential for tumor evasion and are less likely to respond to ICBs. Combining these results with the finding that CRC patients in the high-risk group exhibit lower levels of immune cell infiltration, we believe that our signature has the potential to predict the sensitivity of immunotherapy for CRC patients. However, there is still a long way to go before real clinical application can be achieved.
- The paper does not provide a sufficient comparison of its findings with existing literature. This makes it difficult for readers to assess the novelty and significance of the study's contributions.
Response: We have included three previous studies of cuproptosis in CRC. And we also discussed their limitations. We believe this could help readers understand the novelty and strength of our study.
The relevant description has been added in lines 640-645.
- While the paper claims that the findings may be useful for predicting outcomes and treatment sensitivity in clinical practice, there is no discussion of how this might be achieved or what implications the findings have for patient care.
Response: In revised manuscript, we have discussed about how it could be used as potential clinical applications of CRC:For instance, these genes may serve as biomarkers for detecting CRC at an early stage. And just like the clinical application targeting ferroptosis[60], selectively induction of cuproptosis may also be adopted as a potential treatment strategy in CRC.
The relevant description has been added in lines 655-658.
Thank you very much for your consideration. I look forward to hearing from you.
Sincerely,
Dr. Liying Song
The Third Xiangya Hospital of Central South University

Round 2
Reviewer 1 Report
All comments have been addressed.
Reviewer 2 Report
No further comments to authors. Thank you!